# Crystal structure of tripartite-type ABC transporter MacB from *Acinetobacter baumannii*

Ui Okada [1], Eiki Yamashita[2], Arthur Neuberger [3], Mayu Morimoto[1], Hendrik W. van Veen [3] & Satoshi Murakami [1]

The MacA–MacB–TolC tripartite complex is a transmembrane machine that spans both plasma membrane and outer membrane and actively extrudes substrates, including macrolide antibiotics, virulence factors, peptides and cell envelope precursors. These transport activities are driven by the ATPase MacB, a member of the ATP-binding cassette (ABC) superfamily. Here, we present the crystal structure of MacB at 3.4-Å resolution. MacB forms a dimer in which each protomer contains a nucleotide-binding domain and four transmembrane helices that protrude in the periplasm into a binding domain for interaction with the membrane fusion protein MacA. MacB represents an ABC transporter in pathogenic microorganisms with unique structural features.

[1] Department of Life Science, Tokyo Institute of Technology, Nagatsuta, Midori-ku, Yokohama 226-8501, Japan. [2] Institute for Protein Research, Osaka University, Suita, Osaka 565-0871, Japan. [3] Department of Pharmacology, University of Cambridge, Tennis Court Road, Cambridge CB2 1PD, UK. Correspondence and requests for materials should be addressed to S.M. (email: murakami@bio.titech.ac.jp)

Multidrug resistance caused by export proteins is a serious problem in antibiotic treatment of numerous bacterial infections[1]. The envelope of Gram-negative pathogens, such as *Acinetobacter baumannii* and *Pseudomonas aeruginosa*, contains unique tripartite machineries that export noxious compounds from the cell. These machineries are composed of a plasma membrane transporter and outer membrane

porin (OMP) that are connected by a periplasmic adaptor protein[2]. Proton motive force-dependent tripartite multidrug efflux transporters belonging to the resistance-nodulation-cell division (RND) transporter family have been well characterized[3, 4]. ABC family and major facilitator superfamily transporters can also be part of tripartite complexes, and share similar or identical components with RND transporters, but their structures are still

**Fig. 1** Measurement of drug resistance of *E. coli* expressing *A. baumannii* MacA-MacB-TolC or transport inactive mutants. **a** Growth of *E. coli* expressing *Acinetobacter* MacAB-TolC, or combinations of components and mutants, in the presence of various macrolides and AcrB substrates. Data in Table 1 are based on the colony formation shown in this figure. A, MacA; B, native MacB; B(E172Q), transport-inactive E172Q MacB mutant; B(ΔC10), MacB lacking the carboxy terminal CH2; C, TolC. For MIC measurements, cells were grown on agar plates in the presence of dilutions of compounds. Abbreviations of drugs are the same as in Table 1. **b** Expression of MacA-MacB-TolC confers drug resistance on *E. coli* cells in liquid cultures. Relative growth rate of drug-sensitive *E. coli* W3104 ΔacrABΔmacAB harbouring empty pBAD vector (control), or pBAD containing *macAB/tolC* from *A. baumannii* (blue trace), or pBAD with *macA/tolC* only (inactive; red trace) in the presence of (left) 0–2.3 µg ml⁻¹ azithromycin or (right) 0–56.4 µg ml⁻¹ roxithromycin. IC₅₀ values are summarized in Table 2. Each measurement was repeated three times

**Table 1 Measurement of drug resistance of *E. coli* expressing *A. baumannii* MacA-MacB-TolC or transport inactive mutants**

| Component(s) expressed | MIC (µg ml⁻¹) | | | | | | | |
| --- | --- | --- | --- | --- | --- | --- | --- | --- |
| | Macrolide (MacB substrates) | | | | | AcrB substrates | | |
| | EM | CAM | RXM | AZM | SPM | LM | ACR | SDS |
| None (empty vector) | 1.56 | 1.56 | 4.41 | 0.55 | 4.41 | 2.2 | 4.41 | 50.0 |
| MacA-MacB(E172Q)-TolC | 1.10 | 1.56 | 1.56 | 0.39 | 4.41 | 1.10 | 4.41 | 50.0 |
| MacA-MacB-TolC | 2.21 | 2.21 | 6.25 | 0.55 | 6.25 | 3.13 | 4.41 | 50.0 |
| MacA-MacB | 1.56 | 2.21 | 1.56 | 0.28 | 4.41 | 1.10 | 4.41 | 50.0 |
| MacA-TolC | 1.10 | 1.56 | 1.56 | 0.28 | 2.21 | 1.10 | 4.41 | 50.0 |
| MacB-TolC | 1.56 | 1.56 | 1.56 | 0.28 | 4.41 | 1.10 | 4.41 | 50.0 |
| MacA-MacB(ΔC10)-TolC | 0.78 | 1.10 | 1.56 | 0.28 | 2.21 | 1.10 | 4.41 | 50.0 |

EM, erythromycin; CAM, clarithromycin; RXM, roxithromycin; AZM, azithromycin; SPM, spiramycin; LM, leucomycin; ACR, acriflavine; SDS, sodium dodecyl sulfate
The resistance levels are shown as minimum inhibitory concentration (MIC) in µg ml⁻¹determined by the agar dilution method. Values are based on data shown in Fig. 1. Cells expressing the inactive Walker B E172Q MacB mutant were used as a negative control rather than non-expressing cells that lack the expression of foreign protein. Drug resistance is restored by expression of the functional MacA-MacB-TolC tripartite complex. AcrB substrates acriflavine and SDS are not transported by MacA-MacB-TolC. The MacB mutant lacking the carboxy-terminal coupling helix, CH2 is abbreviated as MacB(ΔC10)

**Table 2 IC$_{50}$ values determined in the growth experiments in Fig. 1b with *E. coli* expressing *A. baumannii* Mac and TolC proteins**

| Macrolide | IC$_{50}$ (µg ml⁻¹) | | |
| --- | --- | --- | --- |
| | Control | MacA/TolC | MacAB/TolC |
| Azithromycin | 0.9 | 0.9 | 1.5 |
| Roxithromycin | 21.8 | 23.3 | 39.3 |

The mean concentrations of antibiotic that reduce the relative growth rate by 50% (IC$_{50}$) in three independent experiments are presented in the table

unknown[5, 6]. Crystal structures of ABC transporters have been reported for various organisms for both substrate exporters and importers[7, 8]. In type I bacterial ABC exporters for antibiotics and cytotoxic agents, the minimum functional unit has two transmembrane domains (TMDs) and two cytosolic nucleotide-binding domains (NBDs). One TMD is often fused to one NBD on a single polypeptide that assembles into a homodimer or heterodimer to form a functional unit. The TMDs can contain six transmembrane helices (TMs) per monomer, but two helices out of six are combined with four helices of the second half-transporter, thus enlarging the dimer interface and providing a tight interaction between the monomers during transport. The TMs are extended at the cytosolic side, forming intracellular domains that interact with NBDs via coupling helices (CHs)[9–12].

The tripartite MacA-MacB-TolC transporter in Gram-negative bacterial pathogens including *Escherichia coli*, *Neisseria gonorrhoeae*, *P. aeruginosa*, *Vibrio cholerae*, *Klebsiella pneumoniae*, *Yiersinia pestis* and *A. baumannii*[5, 13] is an important efflux pump that mediates the extrusion of macrolides[14], peptide toxins[15], virulence factors[16], siderophores[17], lipopolysaccharides[18] and protoporphyrins[19]. MacA-MacB-TolC contains similar components as the relatively well-characterized tripartite RND transporters AcrA-AcrB-TolC in *E. coli* and MexA-MexB-OprM in *P. aeruginosa*. The membrane fusion protein (MFP) MacA is homologous to AcrA and MexA, whereas the OMP TolC is shared by both RND and ABC tripartite systems in *E. coli*, and is homologous to OprM[20]. However, the ABC protein MacB is unrelated to AcrB and MexB, which form large trimeric complexes in the plasma membrane, raising questions about the structure and domain organization of MacB in the tripartite efflux pump. Here, we present the crystal structure of MacB, and show that, by analogy to RND transporters, the MacB homodimer

contains two large periplasmic domains (PLDs) that enable the interactions with the other components in this tripartite efflux pump.

## Results

**Overall structure of MacB.** The ABC exporter MacB from *A. baumannii* was functionally expressed in *E. coli* (Fig. 1, Tables 1, 2, Supplementary Figs. 1 and 14). MacB was purified and its crystal structure was solved at 3.4-Å resolution (Fig. 2a–e, Table 3, Supplementary Fig. 2). A non-critical polypeptide segment between position 248 and 259 at the exterior of the NBD was found to be disordered. MacB shares 83% amino-acid sequence similarity (53% identity) with well-studied MacB from *E.coli* (Fig. 3a) and has a domain arrangement similar to yeast PDR5[21] and mammalian ABCG transporters[22] with an N-terminal NBD followed by a C-terminal TMD (Fig. 3b). Two MacB monomers have a biological twofold rotation axis at the centre of the dimer. Each monomer in the dimer has essentially an identical structure (~1.1-Å RMSD in Cα position), and has elongated TMs (TM1 and TM2) protruding about 25 Å at the periplasmic side of the phospholipid bilayer (Fig. 2, Supplementary Fig. 3). The distal end of these TMs forms the PLD (Fig. 3c, Supplementary Fig. 3) that is important for the interaction with the MFP MacA[23]. The interface between the monomers in the MacB dimer is extensive (2188.9 Å²) with major contributions from the TMDs (1053.0 Å², 48%), PLDs (676.1 Å², 31%) and NBDs (417.0 Å², 19%). Although MacB shares structural motifs with other members of the ABC superfamily, its architecture is unique among transporters of known structures (Fig. 2f).

**Nucleotide-binding domain.** MacB was crystallized in the presence of adenosine-5′-(β-thio)-diphosphate (ADPβS) (Fig. 4a, b, Supplementary Fig. 4). The two NBDs are dimerized in a head-to-tail arrangement, and both subunits contribute conserved Walker A, Walker B and ABC signature motifs and residues to form two composite nucleotide-binding sites at the dimer interface. Electron density was observed in the difference Fourier map (Fig. 4a, b) at the expected position for the nucleotide-binding site. Although the shape of the difference Fourier map is not optimally fitted by the ADPβS model, the phosphor and sulphur atoms of the bound ADPβS are clearly identified in the anomalous difference Fourier map (7σ, 0.013 electron Å⁻³) (Fig. 4a, b, Supplementary Fig. 5). The binding of the adenine moiety of the nucleotide involves a π-stacking interaction with a conserved aromatic residue (Phe16) in the A-loop of the NBD (Fig. 4b, Supplementary Fig. 4)[24]. When compared with crystallized nucleotide-bound and apo states of various ABC exporters

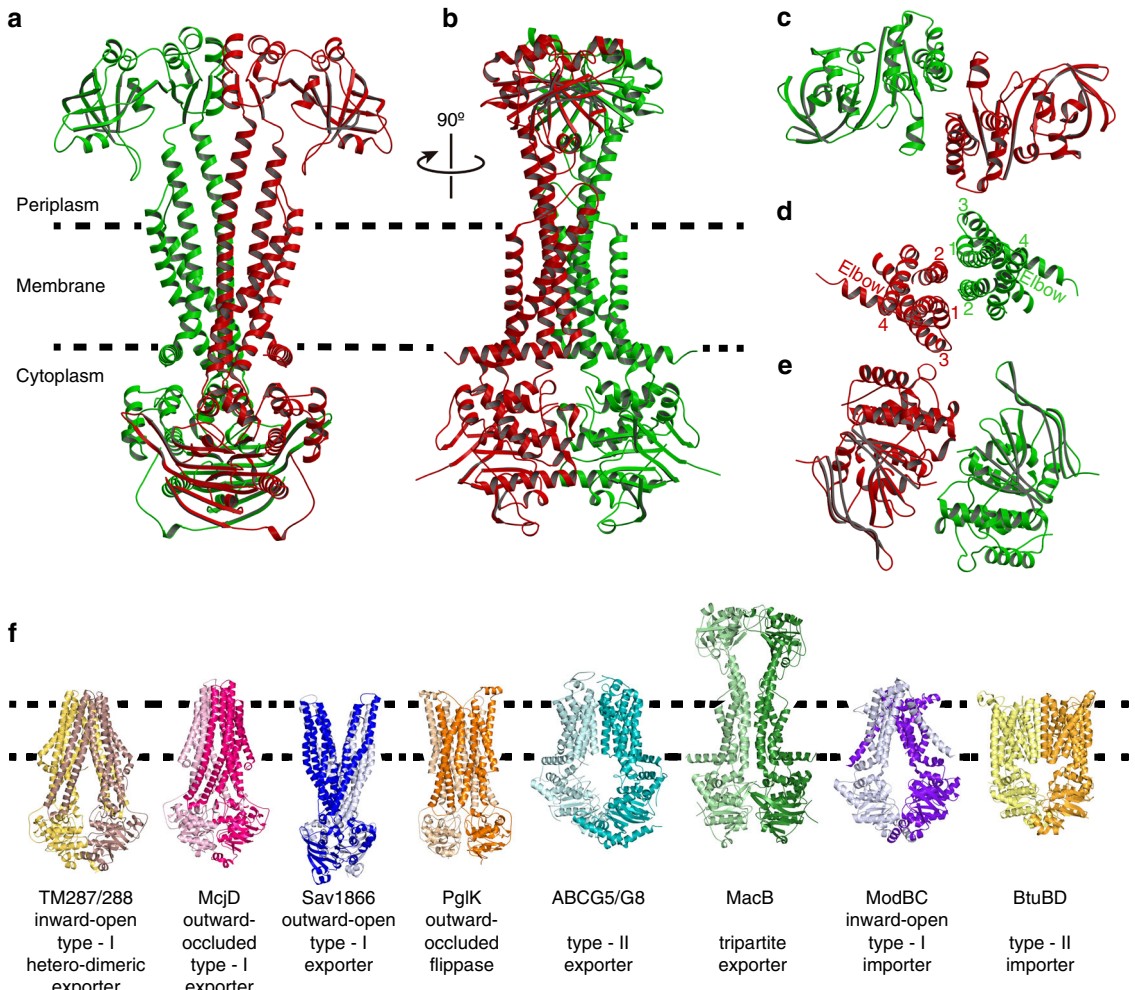

**Fig. 2** Crystal structure of MacB in ribbon representation. **a**, **b** MacB homodimer viewed parallel to the membrane plane in two orientations, rotated by 90° around a vertical axis to the membrane plane. The two protomers are individually coloured (red and green). Dotted lines depict the approximate membrane boundaries on the basis of the hydrophobicity of the protein surface and position of the elbow helix (Supplementary Fig. 3). **c** Top view on PLD domains from periplasmic space. **d** Cut view of the TMD parallel to the membrane plane. TMs are numbered. **e** Bottom view on NBDs from cytoplasm. **f** Comparison with major classes of ABC transporters in different states shows that the MacB structure (in green) is unique. From left to right: type-I (B-family) exporter TM287/288 heterodimer from *Thermotoga maritima* in inward-open state (PDB accession code 3QF4)[11], type-I (B-family) exporter McjD from *E. coli* in outward-occluded state (PDB accession code 4PL0)[12], type-I (B-family) exporter Sav1866 from *Staphylococcus aureus* in outward-open state (PDB accession code 2HYD)[9], lipid-linked oligosaccharide flippase PglK from *Campylobacter jejuni* in outward-occluded state (PDB accession code: 5C73)[25], type-II exporter human ABCG5/ABCG8 (PDB accession code: 5DO7)[26], MacB from *A. baumannii* (PDB accession code: 5WS4), type-I importer ModBC from *Archaeoglobus fulgidus* (PDB accession code: 2ONK)[55] and type-II importer BtuCD from *E. coli* (PDB accession code: 2QI9)[56]. Half-transporters are shown in dark and pale colours to highlight the exchange of TMs in some ABC transporters, but not in the MacB dimer

(Fig. 4c, Supplementary Table 1), the arrangement of the two NBDs in the MacB dimer is most similar to the arrangement of the NBDs in the ADP-bound outward-occluded state of the lipid-linked oligosaccharide flippase PglK[25].

**Transmembrane domain structure**. The TMD of MacB encompasses four TMs per monomer, which is the smallest number among currently characterized ABC exporters[7]. The TMs are not swapped between the half-transporters (Fig. 2f) and lack the helical extensions into intracellular domains as found in most type-I ABC exporters. MacB shares this property with ABC importers and the type-II exporter ABCG5/G8[26] and ABCG2[27]. Even though the MacB monomer has only one cytoplasmic loop, MacB preserves two CHs as observed in other ABC exporters. CH1 is present in the intracellular loop between TM2 and TM3, whereas CH2 is located in the carboxy terminus of MacB (Fig. 3b). Unlike Sav1866 and MsbA, in which CH1 contacts the

NBDs of respective monomers and CH2 interacts with the NBD of the opposite monomer exclusively[9, 10], the two CHs in MacB only contact one NBD within the same monomer. However, even though the CHs in MacB are topologically different and the arrangement of connected TMs shares no similarity with any other bacterial exporter, the placement of the two CHs on the NBD surface is quite similar to the arrangement found in crystallized ABC exporters (Fig. 5, Supplementary Fig. 6, Supplementary Movie 1).The positions of the CHs and their interactions with the NBDs are similar, but the connecting TMs are completely different. Furthermore, the order of the CHs is opposite due to the topological differences (Fig. 5b–d, Supplementary Fig. 6). Removal of CH2 by truncation of the carboxy terminus of MacB leads to inactivation of drug export (Fig. 1a, Table 1, Supplementary Figs. 7 and 14), demonstrating the importance of the CHs for functionality. Furthermore, a loop connects the TMD at its N-terminus to the NBD. This loop contains an amphiphilic

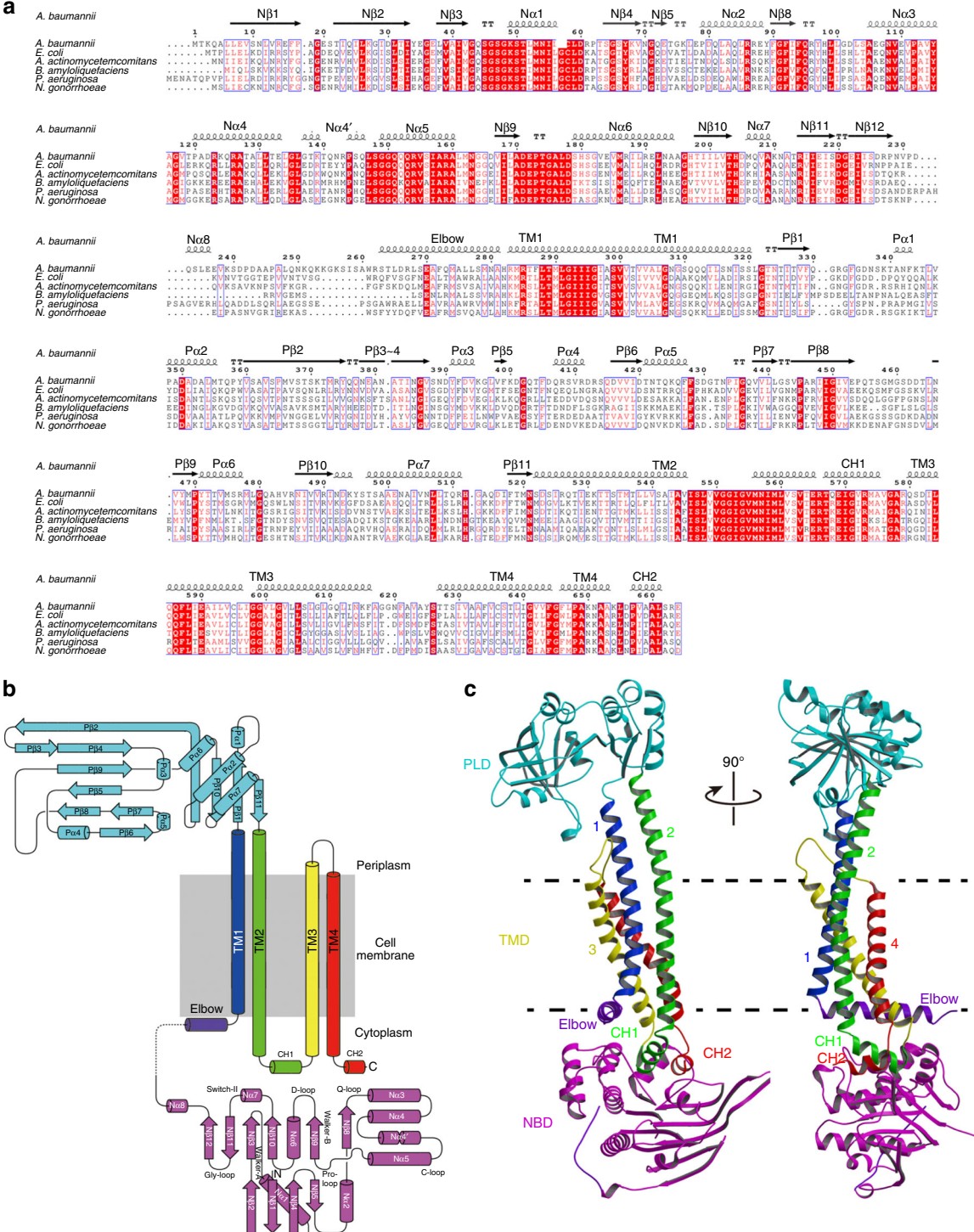

**Fig. 3** Sequence conservation among MacB proteins, secondary and tertiary structures of *Acietobactor* MacB. **a** Amino acid sequence alignment of MacB from *A. baumannii* with five MacB orthologues from *Actinobacillus actinomycetemcomitans* (*Aggregatibacter actinomycetemcomitans*), *Bacillus amyloliquefaciens*, *E. coli*, *P. aeruginosa* and *N. gonorrhoeae*. Alignment was generated with the ClustalW multiple sequence alignment tool[57]. Strictly conserved and highly conserved residues are highlighted in red boxes and red lettering, respectively. Major secondary structure elements are shown on top as coils and arrows for α-helices and β-strands, respectively. Labelling of the secondary structure elements in NBD are according to convention[58]. Alignment was displayed using ESPript 3.0 (http://espript.ibcp.fr/ESPript/ESPript/)[59]. **b** Topology diagram of MacB monomer. Secondary structure elements are indicated with the same annotation as in (**a**). Dotted line shows a disordered polypeptide segment at position 248–259. Designated motifs in NBD are indicated. N and C refer to amino- and carboxy-terminus, respectively. Walker-A, Walker-B, C-loop, D-loop, Q-loop, Gly-loop and Switch-loop are conserved sequence elements in ABC NBDs. **c** Monomer structure of MacB in two orientations, rotated by 90° around a vertical axis to the membrane plane. In **b** and **c**, following colouring is used: NBD in magenta; TMD containing TM1, TM2, TM3 and TM4 in blue, green, yellow and red, respectively; PLD in cyan; elbow in purple. Coupling helix 1 (CH1) between TM2 and TM3 in green. C-terminal CH2 in red. Dotted black lines depict the approximate membrane boundaries on the basis of the hydrophobicity of the protein surface and location of elbow helix. TMs, transmembrane helices; Nα and Nβ, helices and strands in NBD; Pα and Pβ, helices and strands in PLD; Elbow elbow helix, CH, coupling helix

**Table 3 Data processing and refinement statistics**

|  | ADPβS | Se-Met |
|---|---|---|
| Data collection |  |  |
| Space group | $P4_12_12$ | $P4_12_12$ |
| Cell dimensions $a = b, c$ (Å) | 229.7, 154.6 | 229.2, 155.6 |
| Wavelength | 1.7500 | 0.9787 |
| Resolution (Å)[a] | 3.40 (3.46–3.40) | 3.40 (3.46–3.40) |
| $R_{merge}$ | 0.086 (>1.000) | 0.091 (>1.000) |
| $<I> / <\sigma (I)>$ | 27.5 (1.0) | 28.2 (2.0) |
| $CC_{1/2}$ | (0.651) | (0.832) |
| Completeness (%) | 99.9 (100.0) | 99.6 (100.0) |
| Redundancy | 16.3 (15.8) | 8.2 (8.4) |
| Refinement |  |  |
| Resolution (Å) | 3.40 | 3.40 |
| No. of reflections | 46,498 | 57,248 |
| $R_{work}/R_{free}$ | 0.2270 / 0.2571 | 0.2172 / 0.2500 |
| No. of atoms |  |  |
| Protein | 9770 | 9770 |
| Ligand (ADPβS) | 54 | — |
| B factor |  |  |
| Overall | 62.2 | 160.3 |
| R.m.s. deviations |  |  |
| Bond length (Å) | 0.004 | 0.005 |
| Bond angle (degree) | 0.819 | 0.888 |

[a]Values in parentheses are for the highest resolution shell

α-helix running parallel to the cytosolic surface of the plasma membrane in which it is partially embedded (Fig. 3b, Supplementary Fig. 3). This helix is equivalent to the 'elbow helix' in many other ABC exporters, and is also observed in a similar location as the 'connecting helix' in the heterodimeric ABCG-type exporter ABCG5/ABCG8 (Fig. 2f)[26]. Finally, the PLD of MacB has a similar overall structure as the crystallized isolated PLD derived from *Actinobacillus* MacB (r.m.s. deviation of Cα-positions of 2.2 Å over 214 residues)[28] and YknZ, a homologue from *Bacillus amyloliquefaciens* (r.m.s. deviation of Cα-positions of 2.3 Å over 187 residues)[29], except for one loop region containing α-helix, Pα1 (Supplementary Fig. 8). The variable structure of the loop containing Pα1 might point to molecular movements during catalytic activity at the NBDs, and propagation of these movements to MacA and TolC in the transport process. Given the differences in the number of TMs and their interactions in MacB compared to other ABC transporters, and the presence and structure of periplasmic extensions of TM1 and TM2 forming the PLDs, we conclude that the fold of MacB is unique among known structures of ABC exporters (Fig. 2f).

When MacB is compared with outward-facing structures of MsbA and Sav1866, which participate in alternating access mechanisms for these transporters, the overall structure provides a similar V-shape opening with access to the periplasm (Fig. 6, Supplementary Movie 2). However, the locations of these openings are different. The cavity in Sav1866 is present in the outer leaflet of the phospholipid bilayer, whereas the cavity in MacB is present in the periplasmic space. The TMD of MacB does not exhibit a distinct cavity in the membrane-embedded sections of the TMs (Supplementary Fig. 9), suggesting that a substrate-binding pocket might not exist in this part of MacB. This notion is further corroborated in a search of structural similarities between the TMD of MacB and other membrane proteins using the fold match program DALI[30, 31]. Surprisingly, the first 30 significant hits all correspond to ABC transporters. In particular, the arrangement of TM1, 2 and 3 (H282-L321, N521-567T and A578-F616) at the dimer interface of the MacB dimer shows a similarity with the organization of TM1, 2 and 3 (V33-T211) in the mouse multidrug resistance P-glycoprotein

ABCB1a (PDB accession code: 4M1M)[32] (Fig. 7, Supplementary Table 2). In the DALI pairwise comparison, the resulting Z-score of 8.0 (RMSD = 4.1) suggests that these regions share a similar protein structure motif (Fig. 7)[31]. The dimer interface in MacB is not matched with the dimer interface of ABCB1a. Instead, TM1, 2 and 3 in ABCB1a are located at the side of the transporter in a region that faces the phospholipid bilayer, and that is important for conformational changes during transport (Fig. 7c). Similarly, MacB's TM1, 2 and 3 are not part of a substrate-binding cavity in the membrane (Supplementary Fig. 9), and might therefore play an analogous role in energy transduction rather than in substrate interactions. In agreement with this view one of MacB's endogenous substrates, the heat stable enterotoxin II, is matured from precursors in the periplasm and therefore most likely extruded by MacA-MacB-TolC from the periplasm rather than membrane or cytosol[15]. It is also noteworthy that the MacB PLD shares structural similarity with the PLD LolE in the lipoprotein transport system LolCDE (Z = 15.7, RMSD = 3.8) (Supplementary Figs. 10–12)[33], which transfers lipoproteins from the outer leaflet of the inner membrane to the periplasmic carrier protein LolA[33]. Furthermore, MacB PLD is similar to the PLD of AcrB (Z = 7.8, RMSD = 9.0)[28] for which substrate transport from the periplasm has been demonstrated[2] (Supplementary Fig. 13). Thus, MacB might capture substrates from the periplasm. Further structural information about inward-facing states might provide clues regarding the possibility that MacB can also transport substrates from the cytoplasm.

## Discussion

In the AcrAB-TolC and MexAB-OprM assemblies, the proton motive force-dependent functional rotation of the trimeric AcrB and MexB components at the inner membrane is tightly coupled to conformational changes in the PLDs, which in turn drive periplasmic transport of substrate via AcrA/MexA and TolC/OprM[34]. In MacB, the tight coupling between NBD and TMD via CH1 and CH2 will not only be used for movement of TMs but will also be critical for long-range communication with the PLDs that are required for transport of substrate via MacA and TolC. The significant differences in size of the trimeric AcrB and MexB vs.dimeric MacB in otherwise homologous tripartite assemblies raise interesting questions about the stoichiometry of the components in the MacA-MacB-TolC complex[35]. Given the diversity in size and chemical properties of MacB substrates, it will also be interesting to study the mechanisms of substrate binding and transport by the transporter complex. Our crystal structure provides a framework for these and other studies on MacB and related ABC exporters in Gram-negative pathogenic bacteria.

## Methods

**Protein preparation**. The *macB* gene of *A. baumannii* was amplified from genomic DNA (ATCC no. BAA-1710D-5) by the polymerase chain reaction using KOD-plus-Neo DNA polymerase (TOYOBO, Japan) with the forward (5′-ggaattccatatgacaaaacaagctttgcttgaagtc-3′) and the reverse (5′-cgggatcct-tattctcgtgatagtgctgcaacag-3′) primers (Supplementary Table 3), and inserted via *Nde*I and *Bam*HI restriction sites into a modified pET-22 (Novagen) expression vector. Using this method, an N-terminal hexa-histidine tag was added for purification by immobilized metal affinity chromatography. The DNA was sequenced to ensure that only intended changes were introduced. The resulting plasmid was transformed into *E. coli* BL21(DE3) (Novagen) for protein expression. The transformants were grown in ten 5-L flasks at 37 °C in the Davis minimal medium[36] supplemented with 0.2% glucose and 0.1% casamino acid. Expression was induced for 3 h by the addition of 0.1 mM isopropyl-β-D-thiogalactopyranoside at an OD$_{610}$ of 0.6. All subsequent procedures were performed at 4 °C unless indicated otherwise. Cells were collected by centrifugation, resuspended in 50 mM Tris (pH 7.0), 0.5 mM Na-EDTA, 1 mM MgCl$_2$, and disrupted three times using a Microfluidizer M-110EH (Microfluidics Corp., NM, USA) at 15,000 psi. Cell debris was removed by low-speed centrifugation at 27,000×*g* for 10 min. To collect membrane fractions, the supernatant was subjected to ultracentrifugation at 145,000×*g* for 1 h, and washed with 5 mM Tris (pH 7.0), 0.5 mM EDTA. The

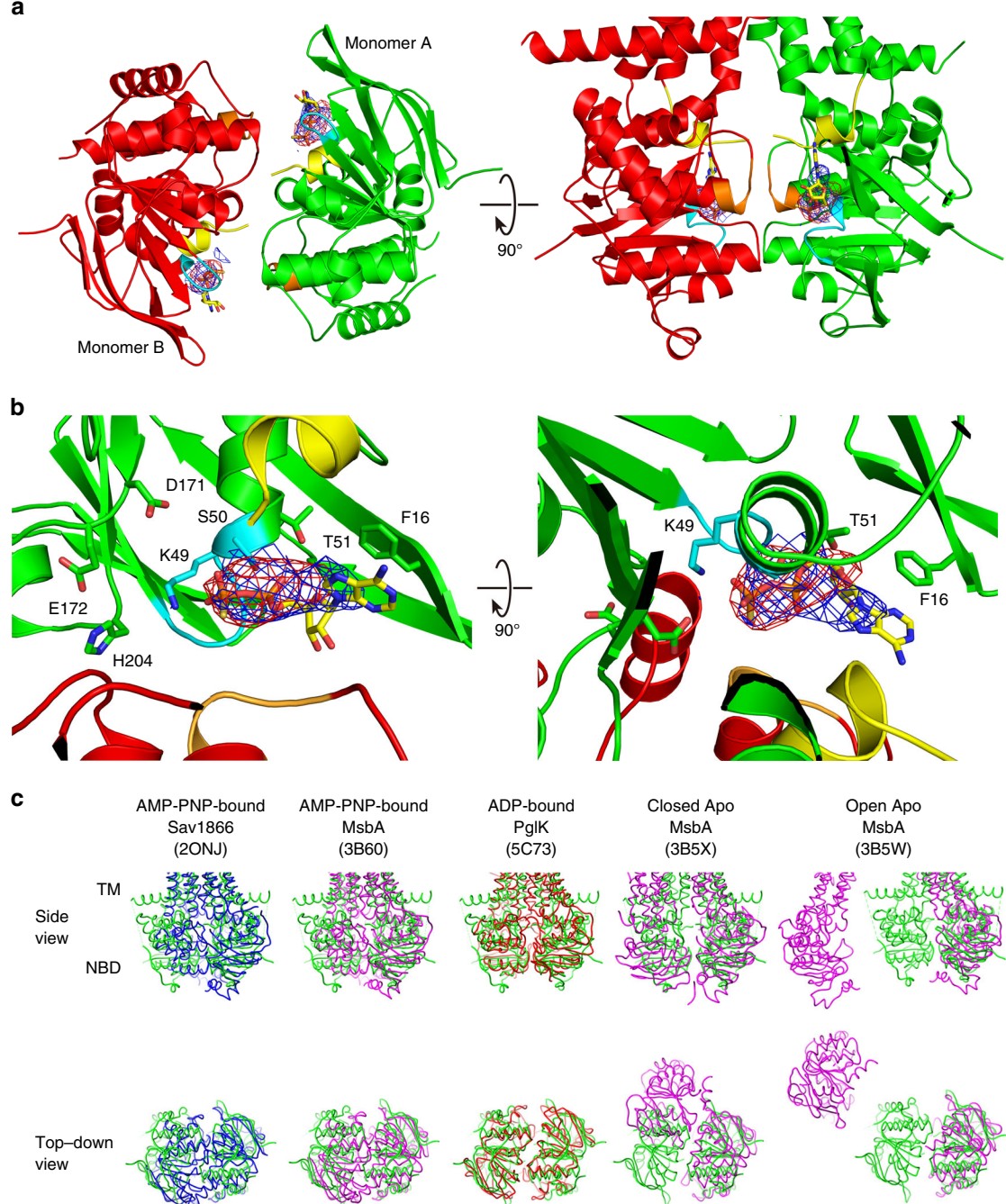

**Fig. 4** Nucleotide-binding sites in MacB dimer. **a** NBD dimer and **b** close-up view of an ATP-binding site interacting with ADPβS. Each monomer in dimeric MacB is coloured in green and red, respectively. Walker A motif in cyan, and carboxy-terminal coupling helix (CH2) in yellow, LSGGQ motif in ABC signature in orange. The difference Fourier map *F*o–*F*c (without ADPβS) contoured at 2σ (σ is the root mean square electron density of the map) (shown in blue cage) and the anomalous difference Fourier map contoured at 4σ (shown in red cage) for phosphor and sulphur atoms of bound ADPβS. Bound ADPβS is shown in stick representation in **a** and **b** with the positions of the C, N, O, P and S atoms indicated by yellow, blue, red, orange and gold, respectively. The positions of key residues of nucleotide-binding site in **b** are also shown in stick representation with the position of the C atoms indicated in green. **c** Structural comparison of NBDs in different nucleotide-bound states. Side view (top row) and top–down view on NBDs (bottom row). Superimposition of the right half of NBD dimers of MacB (green) vs. AMP-PNP-bound Sav1866 from *S. aureus* (PDB accession code: 2ONJ in blue)[60], AMP-PNP-bound MsbA from *Salmonella typhimurium* (PDB accession code: 3B60 in magenta)[10], ADP-bound PglK from *C. jejuni* (PDB accession code: 5C73 in red)[25], closed apo-form MsbA from *V. cholera* (PDB accession code: 3B5X in magenta)[10] or open apo-form MsbA from *E. coli* (PDB accession code: 3B5W in magenta)[10] (Supplementary Table 1). Attempts to superimpose the Mac-NBD on the NBDs of other ABC transporters, such as ABC importers and heterodimeric ABC exporters, were less successful due to differences in the number of helices and the distinct configurations of the NBDs

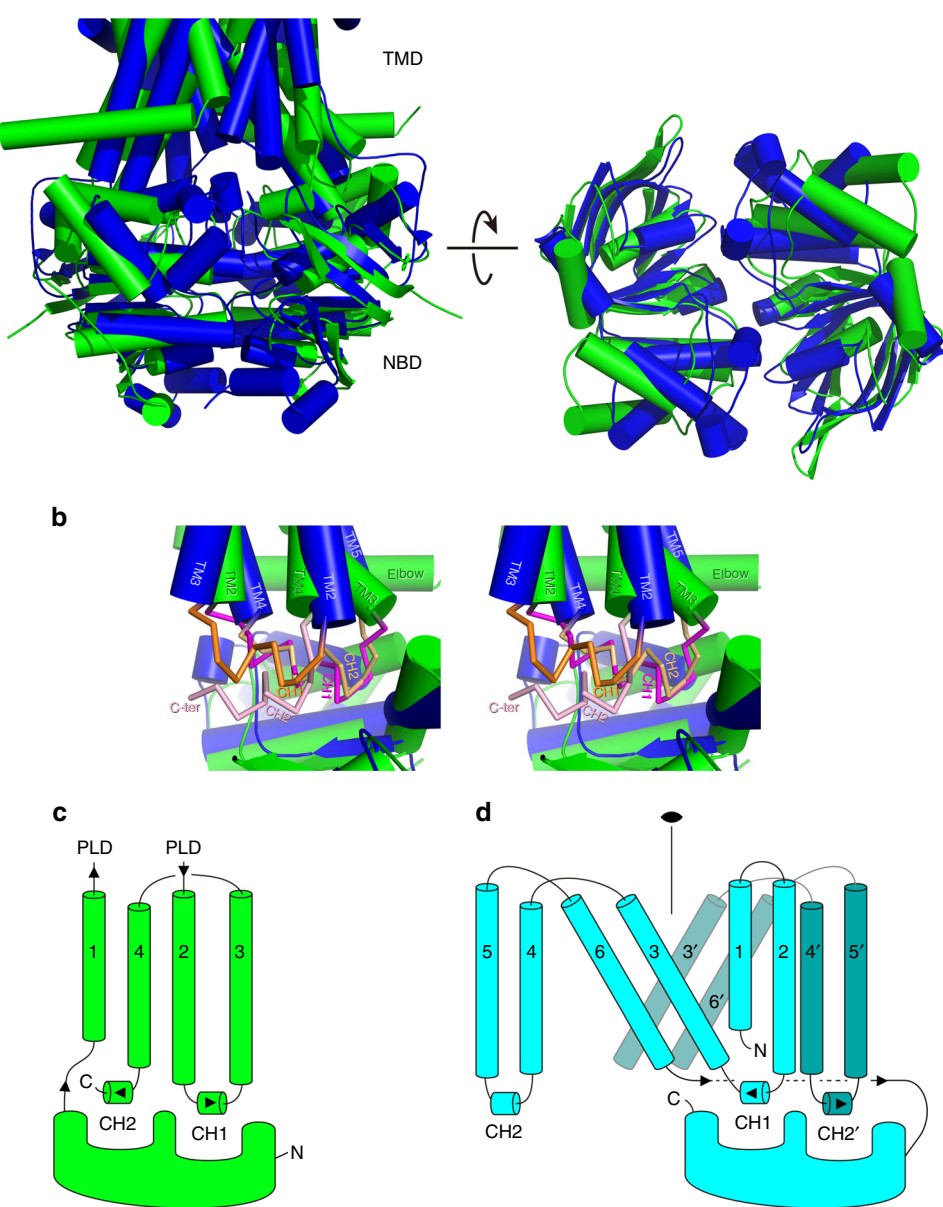

**Fig. 5** Comparison of NBDs and their interactions with coupling helices between MacB and the other type-I ABC exporter. **a** Superimposition of NBDs from MacB and Sav1866 dimer. MacB NBDs (in green) and Sav1866 NBDs (PDB accession code: 2HYD; in blue)[9] are shown in two orientations, rotated by 90° around a parallel axis to the membrane plane, allowing views from (left) the membrane plane and (right) cytoplasm. **b** Monomeric NBD views from the membrane plane are superimposed and shown in stereo-view. MacB and Sav1866 (PDB accession code: 2ONJ)[60] TMs, and elbow and coupling helices, are labelled individually. For NBD and TMD, helices are shown in cylinder representation and CHs are shown in backbone tracing for clarity of presentation. CHs of MacB and Sav1866 are coloured in pink and orange, respectively, and CH1s and CH2s are shown in dark and pale colours, respectively. Sav1866 CH2 is in proximity to MacB CH1, whereas MacB CH2 is close to Sav1866 CH1. **c, d** Topological comparison of CHs and connecting TMs between MacB and other type-I exporters. **c** Topological diagram of MacB monomer coloured green and (**d**) type-I exporter monomer coloured cyan. Neighbouring monomer in the functional dimer in type-I exporter is coloured in dark cyan. Twofold symmetry axis is also shown.

plasma membrane was solubilized in 50 mM Tris (pH 7.0) buffer, 10% (v/v) glycerol containing protease inhibitors (Roche) and 2% (w/v) n-undecyl-β-D-maltoside (UDM, GlyconBiochemicals GmbH, Germany) on ice for 1 h. After a further step of ultracentrifugation at 145,000×g for 1 h, the detergent-solubilized fraction was collected and incubated with chelating sepharose resin immobilized with Ni²⁺ ion at 4 °C for 1 h. The resin was washed with the buffer containing 20 mM Tris (pH 7.5), 100 mM NaCl, 25 mM imidazole, 10% (v/v) glycerol and 0.05% (w/v) UDM. The protein was eluted from the affinity resin with the wash buffer plus 300 mM imidazole. The fractions containing MacB were collected, concentrated in Amicon Stirred Cell (Merck Millipore) with 100 kDa molecular weight cutoff Omega Ultrafiltration Membrane Disc Filter (Pall Corporation, USA), and filtered with Ultrafree-MC GV Centrifugal Filter (Merck Millipore). Further purification was performed by size-exclusion chromatography (Superdex-200 Increase 10/300

GL; GE Healthcare) in the buffer containing 20 mM Tris (pH 7.5), 100 mM NaCl, 10% (v/v) glycerol and 0.05% (w/v) UDM at the flow rate of 0.3 ml min⁻¹ using AKTA explorer 10 S (GE Healthcare). The peak fractions were collected and concentrated in the same way as described above to about 24 mg of MacB per ml for crystallization. Concentration of purified protein was determined by bicinchoninic acid (BCA) protein assay (Thermofisher) with bovine serum albumin as a standard. The purity of MacB was analysed by SDS-PAGE followed by Coomassie Brilliant Blue staining.

**Crystallization**. Adenosine-5′-(β-thio)-diphosphate of 10 mM (ADPβS, Jena bioscience GmbH, Germany) and 2 mM MgCl₂ were added to the purified protein and incubate for overnight on ice before crystallization. MacB crystals were grown

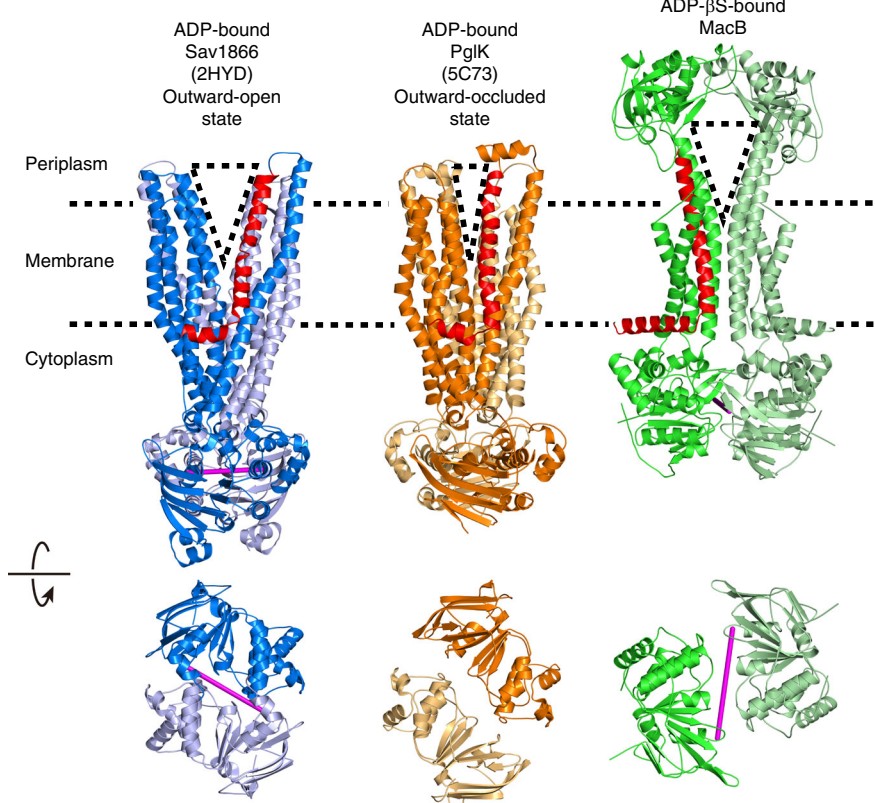

**Fig. 6** Structural comparison between MacB vs. Sav1866 and PglK. Ribbon representation of (left) ADP-bound, outward-open conformation of Sav1866 (blue, PDB accession code: 2HYD)[9], (middle) ADP bound, outward-occluded conformation of PglK (orange, PDB accession code:5C73)[25] and (right) ADPβS-bound MacB (green), viewed from membrane plane (top) and cytoplasm (bottom). Dotted lines depict the approximate membrane boundaries. Alfa phosphate atoms of the two bound nucleotides are connected by a magenta line showing the interface in each NBD dimer. For ADP-bound PglK, the coordinates of ADP were not deposited in the Protein Data Bank[25]. Outward-facing, V-shaped cavities in the TMDs are depicted as dotted triangles. Elbow helix and connecting TM1, which contribute to the extent of the external opening in each transporter, are coloured in red

by the sitting drop vapour diffusion technique at 25 °C. The protein solution was mixed (1:1) with reservoir solution containing 1.2–1.3 M sodium citrate, 100 mM Na-HEPES (pH 7.2). Crystals were grown within 1 week to optimal size (0.3 × 0.3 × 0.2 mm³). The concentration of glycerol was gradually increased to 30% (v/v) by soaking in several steps for optimal cryo-protection. Crystals were picked up using nylon loops (Hampton Research, CA, USA) for flash-cooling in cold nitrogen gas from a cryostat (Rigaku, Japan). For Se-methionine labelling, E. coli cells were grown in M9 minimal medium (L-methionine was replaced with seleno-L-methionine), and the protein was purified and crystallized as described for native protein with the exception of the addition of 2 mM adenosine-5′-triphosphate (ATP, Nacalaitesque, Japan) instead of 10 mM ADPβS before crystallization.

**Data collection and structure determination**. Data sets were collected at 100 K using a Rayonix MX300-HE charge-coupled device detector on the BL44XU beamline at SPring-8. Diffraction images were processed with the HKL2000 package[37]. Further processing was carried out with programs from the CCP4 suite[38] and Phenix[39, 40]. Data collection and structure refinement statistics are summarized in Table 3. Native data was collected at a wavelength of 1.7500 Å. The initial phases were obtained by SAD (the single anomalous dispersion method) using SeMet derivative crystals. The heavy atom sub-structure of 19 selenium atoms was determined by SHELXD, and initial phases were obtained using SHELXE and SHARP[41, 42], and were used to calculate anomalous Fourier maps using programs from the CCP4 suite[38]. Density modifications with solvent flattening and non-crystallographic averaging to phase improvement was performed using DM[43]. Model building was carried out using programs O[44] and COOT[45]. Chain tracing was aided by the known positions of methionines from the selenomethionine data. Model refinement was conducted using Refmac5[46] and Rosetta-Phenix[47] and Phenix[39, 40]. Ramachandran analysis revealed 95.6% in the favoured region and 0.0% residues in the outliers with a Molprobity[48] score of 1.49. The surface area between the MacB monomers was calculated using QT-PISA (CCP4 suite)[49]. Figures were prepared using Molscript[50] rendered with Raster3D[51], Chimera[52] and PyMOL.

**Determination of MIC values**. The expression vector pBAD24 containing macA-macB-tolC genes of A. baumannii, deletion mutants, or E172Q mutant of macB (Supplementary Table 3), were transformed into E. coli strain W3104 ΔacrABΔ-macAB[53]. These expression hosts were additionally transformed with the pRARE2 plasmid, which supplies transfer RNAs for rarely used codons in E. coli. The pRARE2 plasmid was isolated from Rosetta 2 (Novagen). The minimum inhibitory concentrations (MICs) of macrolide antibiotics were determined as the concentrations that prevented bacterial growth after 30 h incubation at 30 °C on YT-agar (0.8% tryptone, 0.5% yeast extract, 0.5% NaCl, 1.5% agar, 100 μg ml⁻¹ ampicillin and 34 μg ml⁻¹ chloramphenicol) plates with sequential dilutions, according to standard protocol recommended by the Clinical and Laboratory Standards Institute[54]. To induce protein expression, 2% L-arabinose was added in the agar plates. Expression of MacA, MacB or TolC in the strains used for MIC measurements was confirmed by western blotting with anti-MacA polyclonal (1:8000 dilution), anti-MacB monoclonal (1:500 dilution) or anti-TolC polyclonal (1:8000 dilution) antibodies as the primary antibody, and horseradish peroxidase (HRP)-labelled goat anti-rabbit (1:20,000 dilution, cat# 111-035-003, Jackson ImmunoResearch, USA) or anti-mouse (1:10,000 dilution, cat# 115-035-062, Jackson ImmunoResearch) IgG antibodies as the secondary antibody.

**Truncation of second coupling helix**. To express MacB without CH2, a stop codon was introduced by replacing the Leu655 codon in the macB gene through the use of an inverse PCR-based site-directed mutagenesis kit using KOD DNA polymerase (TOYOBO) with the two primers (5′-caaaataa-gaccctgttgcagcactatcacgag-3′, 5′-ctgcattcttggcaggtaagaagcc-3′) (Supplementary Table 3) and pBAD24 vector containing macA-macB-tolC genes as a template. The resulting plasmid was sequenced to confirm successful truncation, and codes for a truncated MacB protein lacking the C-terminal 10 residues that correspond to CH2. Expression of this truncated MacB was confirmed by western blotting with anti-MacB polyclonal antibody (1:1000 dilution) and HRP-labelled goat anti-rabbit IgG antibody (1:20,000 dilution, cat# 111-035-003, Jackson ImmunoResearch). The mutant was used for the MIC measurement as well as the wild-type protein.

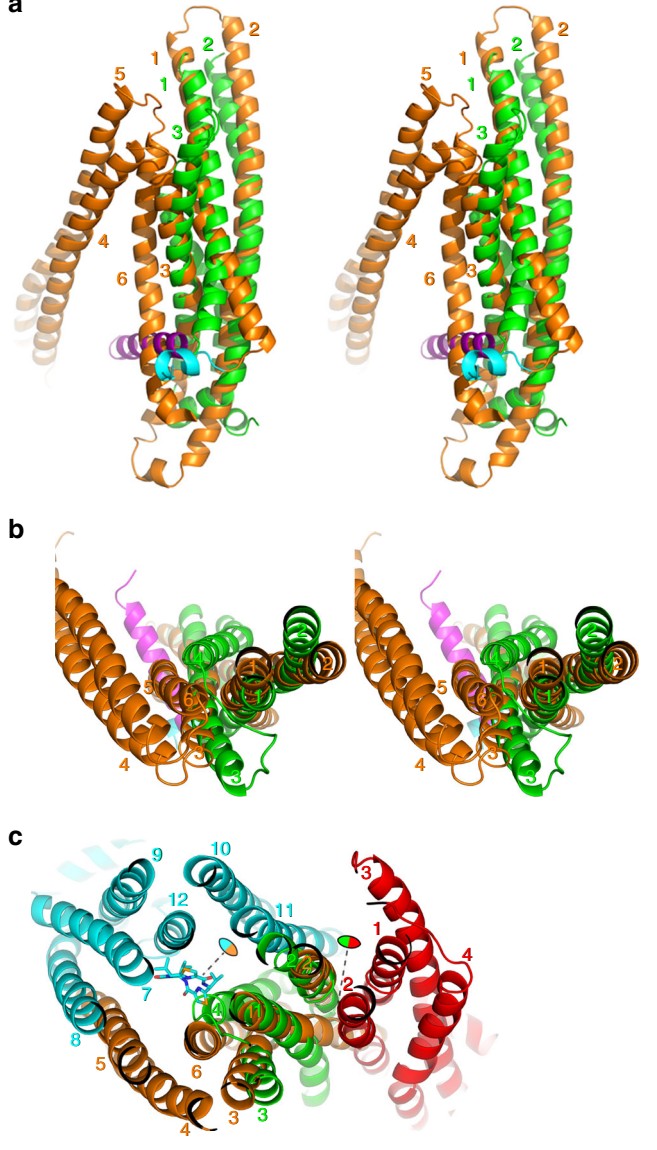

**Fig. 7** Superimposition of helix bundles of MacB and mouse ABCB1a. **a**, **b** MacB monomer and N-terminal half of ABCB1a are superimposed (in stereo view) according to the top hit of the fold match program DALI. MacB (green) and ABCB1a (PDB accession code: 4M1M; orange)[32] are shown in (**a**) side view and (**b**) top view. TMs are numbered. TM1-3 from each transporter are essentially overlapping with high $Z$-score ($z = 8.0$). Elbow helices of MacB and ABCB1a are coloured in magenta and cyan, respectively. **c** Functional unit of MacB and ABCB1 are shown in a similar orientation as in **b**. Two monomers of MacB are coloured in green and red. Two halves of ABCB1a are coloured in cyan and orange. Monomer A (green) in dimeric MacB is superimposed on N-terminal half of mouse ABCB1a (PDB accession code: 4M2S; N-terminal half coloured in orange)[32]. Inhibitor, QZ59-RRR ((4R,11R,18R)-4,11,18-tri(propan-2-yl)-6,13,20-triselena-3,10,17,22,23,24-hexaazatetracyclo [17.2.1.1 ~ 5,8 ~ .1 ~ 12,15~] tetracosa-1(21),5(24),7, 12(23),14,19(22)-hexaene-2,9,16- trione)) in the substrate binding pocket between TM1, TM6, TM7 and TM12 of ABCB1a is indicated in stick representation. Biological twofold symmetric axes on each transporter are shown by dotted lines and coloured ovals

**Measurement of cellular growth rates**. Overnight cultures from glycerol-stocks of drug-hypersensitive *E. coli* Δ*acrAB* Δ*macAB*, harbouring pBAD plasmid encoding *A. baumannii* MacAB-TolC, MacA-TolC (inactive control) or none of these genes (inactive non-expressing control), were grown in Luria-Bertani broth containing

100 µg ml$^{-1}$carbenicillin. Cultures were grown for about 16 h at 37 °C, and then used to inoculate fresh medium (1:1000 dilution). Once *E. coli* cells had reached $OD_{600}$ of 0.3–0.4, protein production was induced by addition of 2% arabinose for 3 h. The cells were then diluted to $OD_{600}$ of 0.06 in fresh medium containing 2% arabinose in the wells of a 96-well plate to which antibiotics were added as indicated in Fig. 1b. Growth was followed over time at $OD_{600}$ at 37 °C in a Versamax microplate reader (Molecular Devices, USA).

**Data availability**. Coordinates and structure factors have been deposited in the Protein Data Bank with accession codes: 5GKO and 5WS4. Data supporting the findings of this study are available within the article and its Supplementary Information files. Data have also been deposited in the University of Cambridge data repository (https://doi.org/10.17863/CAM.12992) or are available from the corresponding author upon reasonable request.

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

## Acknowledgements

This work was supported by a Grant-in-aid in the funding program for Next Generation World-Leading Researchers (NEXT Program) from the Japan Society for the Promotion of Science (JSPS) (to S.M.), Targeted Proteins Research Program from the Ministry of Education, Culture, Sports, Science and Technology of Japan (MEXT) (to S.M. and E.Y.), ERATO "Lipid Active Structure Project" from Japan Science and Technology Agency (JST) (to S.M.) and the Advanced Research for Medical Products Mining Programme of the National Institute of Biomedical Innovation (NIBIO) Japan (to S.M.). We are also grateful for funding by the Strategic International Cooperative Program (JST, Japan) (to S.M. and H.W.v.V.), Royal Society (UK) (to H.W.v.V.), BBSRC (UK) (to H.W.v.V.) and International Human Frontier Science Program Organization (HFSPO) for collaborative research between S.M. and H.W.v.V. A.N. is the recipient of a Herchel-Smith Scholarship. We thank M. Kato and W. Tamaya for technical assistance. The synchrotron radiation experiments were performed at the BL44XU of SPring-8 with the approval of the Japan Synchrotron Radiation Research Institute (JASRI) (proposal no. 2012A6746, 2012B6746, 2013A6855, 2013B6855, 2013B6700, 2014A6954, 2014A6700, 2015A6700 and 2016A6700). This research is partially supported by the Platform Project for Supporting Drug Discovery and Life Science Research (Platform for Drug Discovery, Informatics, and Structural Life Science) from MEXT and Japan Agency for Medical Research and Development (AMED). The pBAD24 vector was provided by the National Institute of Genetics through the National Bio-Resource Project of the MEXT, Japan. This work was done with the approval of the Joint Research Committee of Institute of Protein Research, Osaka University.

## Author contributions

U.O. performed cloning, purification, crystallization and MIC measurements. A.N. performed growth rate analyses, and A.N. and H.W.v.V. analysed the data. M.M. established CH2 truncated mutant. U.O. and S.M. processed crystals for X-ray experiments. Synchrotron experiments were performed by U.O., S.M. and E.Y. Data processing and crystallographic calculation were performed by E.Y. and S.M. Manual model building and refinement were performed by S.M. S.M. supervised the overall project. S.M. and H.W.v.V. wrote the manuscript.

## Additional information

**Competing interests:** The authors declare no competing financial interests.

