## [Peer Review File · Nature Communications]

Reviewers' comments:

Reviewer #1 (Remarks to the Author):

In bacteria tripartite transporters cause resistance to antibiotics by expelling them through a range of tripartite transporters, which are usually composed of a series of three proteins. One of these systems contains the protein MacB, a dimeric ABC transporter. MacB resides in the inner membrane, and is part of a complex with MacA and TolC, that extends between the inner and outer membrane. It expels macrolide antibiotics and other large molecules from bacteria such as *E. coli*, *N. meningitidis* and *A. baumannii*. Up to this point there has been no structure for MacB, the ABC transporter component of this particular tripartite pump system.

This paper describes an X-ray crystallography structure of MacB from *Acinetobacter baumannii*. This is a novel and impactful result. This structure has a novel fold. It has an N-terminal nucleotide binding domain, which has only been seen in one ABC transporter structure, the recently published ABCG5/8 structure. All other ABC transporters have the NBDs on the C-terminus of the chain, between the TMDs or on a separate chain.

Unlike all previously solved ABC transporter structures, MacB has only 4 TM helices per chain. It lacks the two additional helices that form a domain swap between the domains of ABC transporters of the exporter type. There is a remarkable conservation of elements such as the CH1 and CH2 helices, which interact with the NBDs, despite the lack of conservation of the order of structural elements within the protein. CH2 is found at the end of the sequence, and is present on the same chain as the rest of the monomer, rather than being provided by a domain swap of TM4 and TM5. There is also a helix that is equivalent to the "elbow helix" seen in classic ABC exporters, but it is found at the C-terminus of the chain, rather than at the N-terminus, as observed in most ABC transporters. This feature resembles the ABCG5/8 structure.

This structure also features an extension between TM1 and TM2, that goes into the periplasmic space. This structure locks into the MacA structure, connecting the cytoplasm with the tube that forms between 6 MacA molecules, allowing transport substrates to pass directly from MacB to TolC, without being released into the periplasm.

In general the results are novel, unexpected and of great value to the structural biology, drug resistance and bacteriology communities. This work represents a very considerable advance in our understanding of drug resistance related proteins and their mode of action.

The methods used in this paper to solve the structure are standard crystallography, similar to many other membrane protein structures. The approach appears to be valid and the data is of sufficient quality to support most of the conclusions in the paper.

The authors describe in detail the nucleotide bound to the protein. They say that they have confirmed the nature of the nucleotide by showing both sulphur and phosphorus anomalous peaks for ADPbetaS, confirming that it is this nucleotide that is bound. However, the maps shown are still not very convincing. The sulphur and phosphorus atoms should presumably sit in the middle of the anomalous difference peak, since there are no other heavy atoms, but in fact they are not in the centre of the peak, they are displaced to one side of the anomalous peak, at least in figure 2.

Given the signal from the phosphorus atoms, it is not clear that there is additional signal from the sulphur atom. There is very little density for the rest of the nucleotide, even at 2 sigma. Perhaps if they sharpened the maps, they could improve this view. Never-the-less they don't have very good evidence for the nature of the nucleotide bound to the protein, at least from the maps shown.

There is no data on the quality of the anomalous data in the dataset descriptions.

Pg 3 last paragraph: "The TMs are not swapped between the last half-transporters". Are they not swapped, or are the two additional helices that would be swapped in fact simply missing?

Pg 4. Line 16 and 18: The authors state that an rmsd of 4.5A on c-alphas over 185 residues indicates a high level of similarity. Perhaps this should be toned down. 4.5 A is not a very high similarity, over a short section of a protein, including only the C-alphas. "Distantly related" might be more reasonable for this comparison.

Pg. 5 first paragraph. The discussion of the substrate binding sites and comparison with Pgp is not entirely convincing. These two proteins are not at all closely related, the folds are in fact rather dis-similar, number of helices, length of connections between the transmembrane region and the

NBDs, the extracellular extensions are all different. Given these differences, I would not expect that the substrate binding site would be conserved to a noticeable extent. Therefore the observation that the regions of similarity are on the outside of Pgp, does not convince me that this conformation does not have a substrate binding site in the cavity, or that the substrate binding site was on the surface. I would not expect the substrate binding site to be detectable through homology with Pgp. This section should perhaps be re-written.

In order to identify the conserved regions in MacB, it would be a great deal more informative to produce a large sequence alignment with 60 different organisms and look for more and less well conserved patches. If the surface of the protein is generally well conserved, perhaps it is the substrate binding site, if it is not conserved, then this is less likely. Of course if the nature of the substrate changes, then the nature of the binding site will not be conserved, so the homologues should be selected carefully. There are two sequence alignments in the supplementary figures section, which would be a good starting point for this comparison.

Also the relationship to the heat stable enterotoxin II transport is not very clear. Are they suggesting that the enterotoxin binds to MacB on the same site as the conserved Pgp homology, which is inside the membrane? This seems unlikely. If it binds to the periplasmic domain, that would be considerably more convincing.

The suggestion that some substrates are captured from the periplasm seems reasonable/possible, the relationship with Pgp seems less clear. In fact there is little supporting evidence for either view.

Figure 1. The authors claim a similarity to Pgp, so why is Pgp not included in the alignment of structures? It is in fact a very different looking protein, so

Supplementary figure 3 – This figure is very confusing. Does it not show that there is no relationship between the two structures? Topology and arrangement of helices are completely different.

Suppl. Figure 18 – it seems highly unlikely that this alignment is correct. If the 4 helices aligned to not recapitulate the dimer in any way, then it is probable that the alignment is spurious. The structures compared are very different. The alignment as shown is improbable.

Suppl. Figure 20 – “similar structural motifs” not motives perhaps.

Why two sequence alignments, Suppl. Figure 1 and Suppl. Figure 21 both show alignments to related proteins. Could these not be combined?

Reviewer #2 (Remarks to the Author):

This is an excellent study describing the crystal structure of MacB transporter. This structure is the first of its kind and provides novel exciting insight into the family of ABC transporters about which we have very limited knowledge. The manuscript is concise and clearly written and will be of interest to a broad scientific community.

The previous criticism is addressed constructively.

A minor correction:

1. Fig S5 is the only place where NodT protein is mentioned in the results. Would be helpful to if some explanations are provided.

Reviewer #3 (Remarks to the Author):

The authors have fully addressed my previous comments. This crystal structure was critical to interpret the cryo-EM reconstruction of the MacA-MacB-ToIC complex that was recently published in Nature Microbiology. I recommend publication without any delay.

Reviewer #4 (Remarks to the Author):

The authors have sensibly removed the unduly speculation from their previous version of the manuscript. I have only two points left:

- I would like to see the PDB validation report, for quality check on the crystal structure.
- The authors should provide an omit map for figure 2b.

Reviewers' comments:

Reviewer #1 (Remarks to the Author):

In bacteria tripartite transporters cause resistance to antibiotics by expelling them through a range of tripartite transporters, which are usually composed of a series of three proteins. One of these systems contains the protein MacB, a dimeric ABC transporter. MacB resides in the inner membrane, and is part of a complex with MacA and TolC, that extends between the inner and outer membrane. It expels macrolide antibiotics and other large molecules from bacteria such as E. coli, N. meningitidis and A. baumannii. Up to this point there has been no structure for MacB, the ABC transporter component of this particular tripartite pump system.

This paper describes an X-ray crystallography structure of MacB from Acinetobacter baumannii. This is a novel and impactful result. This structure has a novel fold. It has an N-terminal nucleotide binding domain, which has only been seen in one ABC transporter structure, the recently published ABCG5/8 structure. All other ABC transporters have the NBDs on the C-terminus of the chain, between the TMDs or on a separate chain.

Unlike all previously solved ABC transporter structures, MacB has only 4 TM helices per chain. It lacks the two additional helices that form a domain swap between the domains of ABC transporters of the exporter type. There is a remarkable conservation of elements such as the CH1 and CH2 helices, which interact with the NBDs, despite the lack of conservation of the order of structural elements within the protein. CH2 is found at the end of the sequence, and is present on the same chain as the rest of the monomer, rather than being provided by a domain swap of TM4 and TM5. There is also a helix that is equivalent to the 'elbow helix' seen in classic ABC exporters, but it is found at the C-terminus of the chain, rather than at the N-terminus, as observed in most ABC transporters. This feature resembles the ABCG5/8 structure.

This structure also features an extension between TM1 and TM2, that goes into the periplasmic space. This structure locks into the MacA structure, connecting the cytoplasm with the tube that forms between 6 MacA molecules, allowing transport substrates to pass directly from MacB to TolC, without being released into the periplasm.

In general the results are novel, unexpected and of great value to the structural biology, drug resistance and bacteriology communities. This work represents a very considerable advance in our understanding of drug resistance related proteins and their mode of action.

The methods used in this paper to solve the structure are standard crystallography, similar to many other membrane protein structures. The approach appears to be valid and the data is of sufficient quality to support most of the conclusions in the paper.

Our response:

We appreciate Referee 1's great evaluation.

The authors describe in detail the nucleotide bound to the protein. They say that they have confirmed the nature of the nucleotide by showing both sulphur and phosphorus anomalous peaks for

ADPbetaS, confirming that it is this nucleotide that is bound. However, the maps shown are still not very convincing. The sulphur and phosphorus atoms should presumably sit in the middle of the anomalous difference peak, since there are no other heavy atoms, but in fact they are not in the centre of the peak, they are displaced to one side of the anomalous peak, at least in figure 2. Given the signal from the phosphorus atoms, it is not clear that there is additional signal from the sulphur atom.

Our response:

Thank you for the comments. Because of the limited resolution at 3.4 Å, the quality of the electron density map cannot not reach its perfect fit. We therefore added new experimental electron density (2mFo-DFc) data as Supplementary Fig. 4 just to show the true quality of bound nucleotide's electron density. It is truly 3.4 Å level as Referee#1 mentioned. Thus, we conducted an X-ray experiment with low energy X-ray (@1.75 Å) to observe anomalous dispersion from sulfur and/or phosphor atoms in order to prove that the interesting electron density at the expected position in the nucleotide-binding site is indeed ADPβS. As we shown in Supplementary Fig. 5, we can observe all the sulfur atoms (except the first Met), and the anomalous peaks of phosphor and sulfur of the ADPβS are exactly overlapping with the electron density in the nucleotide binding site (2mFo-DFc : supplementary Fig. 4 and Fig. 2a,b).

There is very little density for the rest of the nucleotide, even at 2 sigma. Perhaps if they sharpened the maps, they could improve this view. Never-the-less they don't have very good evidence for the nature of the nucleotide bound to the protein, at least from the maps shown.

There is no data on the quality of the anomalous data in the dataset descriptions.

Our response:

Thank you for the comments. We (have already) refined both native and anomalous data (Low energy X-ray at 1.75 Å) and shown their statistics in Table 3. The quality of the electron density map is in agreement with this resolution. However, distinct anomalous signals can be observed and are shown for sulfur and phosphor in the Supplementary Fig. 5.

Pg 3 last paragraph: "The TMs are not swapped between the last half-transporters". Are they not swapped, or are the two additional helices that would be swapped in fact simply missing?

Our response:

Thank you for the comments. This sentence no longer exists in the revised version. We feel that we cannot comment in a reliable way on the topology of additional TMs which are absent in our MacB structure.

Pg 4. Line 16 and 18: The authors state that an rmsd of 4.5A on c-alphas over 185 residues indicates a high level of similarity. Perhaps this should be toned down. 4.5 A is not a very high similarity, over a short section of a protein, including only the C-alphas. "Distantly related" might be more reasonable for this comparison.

Our response:

Thank you very much for the comment. As referee#1 mentioned the RMSD was high indeed, despite the fact that our MacB PLD (325-522), *Actinobacillus* MacB PLD (PDB ID: 3FTJ) and YknZ PLD, a homologue from *Bacillus* (PDB ID: 5F9Q) are genuine orthologues. We conducted further structural comparisons of these domains using the DALI server, and can now correct the RMSD to 2.2 and 2.3, which are reasonable for the C α displacement between homologous proteins with small structural changes. We changed the RMSD values in the revised manuscript. We are so grateful for this reminder of referee#1's.

-----Cut and paste from the result output

Query: MACB PLD(325 ~ 522)

No: Chain Z rmsd lali nres %id PDB Description

1: 3ftj-A 24.5 **2.2** 182 214 3 PDB MACROLIDE EXPORT ATP-BINDING/PERMEASE PROTEIN

2: 5f9q-A 20.7 **2.3** 174 187 6 PDB MACROLIDE EXPORT ATP-BINDING/PERMEASE PROTEIN YKN

Pg. 5 first paragraph. The discussion of the substrate binding sites and comparison with Pgp is not entirely convincing. These two proteins are not at all closely related, the folds are in fact rather dissimilar, number of helices, length of connections between the transmembrane region and the NBDs, the extracellular extensions are all different. Given these differences, I would not expect that the substrate binding site would be conserved to a noticeable extent. Therefore, the observation that the regions of similarity are on the outside of Pgp, does not convince me that this conformation does not have a substrate binding site in the cavity, or that the substrate binding site was on the surface. I would not expect the substrate binding site to be detectable through homology with Pgp. This section should perhaps be re-written.

In order to identify the conserved regions in MacB, it would be a great deal more informative to produce a large sequence alignment with 60 different organisms and look for more and less well conserved patches. If the surface of the protein is generally well conserved, perhaps it is the substrate binding site, if it is not conserved, then this is less likely. Of course, if the nature of the substrate changes, then the nature of the binding site will not be conserved, so the homologues should be selected carefully. There are two sequence alignments in the supplementary figures section, which would be a good starting point for this comparison.

Our response:

We entirely agree with the reviewer and apologize for the lack in clear explanation from our side. We did not predict or suggest the location of the substrate binding pocket of MacB in our manuscript through comparison with ABCB1a. We actually show that the substrate binding pocket

cannot be predicted by comparison with ABCB1, because MacB and ABCB1a have different structures at the quaternary structure level. But these two proteins share similar STRUCTURAL MOTIFS demonstrated by the DALI search. With these STRUCTURAL MOTIFS, we refer to a local similarity of a small structural part. We never discussed a similarity over entire structures. To make this point clear we have rewritten the relevant sections in the main text of the manuscript with additional DALI calculations. Furthermore, we have redrawn all the figures related to this topic (Fig. 7). To provide additional information about the DALI algorithm and the Z-score we added the reference (Ref. 31): Holm, L. *et al.* Using Dali for structural comparison of proteins. *Curr Protoc Bioinformatics*. Chapter 5, Unit 5 5 (2006).

Also the relationship to the heat stable enterotoxin II transport is not very clear. Are they suggesting that the enterotoxin binds to MacB on the same site as the conserved Pgp homology, which is inside the membrane? This seems unlikely. If it binds to the periplasmic domain, that would be considerably more convincing.

The suggestion that some substrates are captured from the periplasm seems reasonable/possible, the relationship with Pgp seems less clear. In fact there is little supporting evidence for either view.

Our response:

There was another point of confusion for which we apologize, and we have rewritten the main text to clarify this point. MacB transports very large substrates such as heat-stable enterotoxin II, which is matured from precursors in the periplasm. This suggests that the binding of this substrate by MacB occurs in the periplasm.

Figure 1. The authors claim a similarity to Pgp, so why is Pgp not included in the alignment of structures? It is in fact a very different looking protein, so

Our response:

We show Sav1866 as a representative of Type-I exporter, rather than ABCB1. This figure (newly renumbered as Fig 2f) got too tight and small with 8 different types of ABC proteins. Therefore, ABCB1 was omitted for reader-friendly reasons.

Supplementary Figure 3 -; This figure is very confusing. Does it not show that there is no relationship between the two structures? Topology and arrangement of helices are completely different.

Our response:

We removed Supplementary Fig. 3 from the revised version.

Suppl. Figure 18 - it seems highly unlikely that this alignment is correct. If the 4 helices aligned to not recapitulate the dimer in any way, then it is probable that the alignment is spurious. The structures compared are very different. The alignment as shown is improbable.

Suppl. Figure 20 - "similar structural motifs" not motives perhaps.

Our response:

Thank you for the suggestion which is similar to the suggestion of Referee#1. In the revised version of our manuscript, these points are corrected in Fig. 7, and associated legends.

Why two sequence alignments, Suppl. Figure 1 and Suppl. Figure 21 both show alignments to related proteins. Could these not be combined?

Our response:

Thank you for the comment. It might seem to be superfluous. However, the former alignment is to explain strict conservation within the MacB family including *A. Baumannii* MacB. And the latter alignment is to compare two related protein families to which MacB and LolE belong.

Reviewer #2 (Remarks to the Author):

This is an excellent study describing the crystal structure of MacB transporter. This structure is the first of its kind and provides novel exciting insight into the family of ABC transporters about which we have very limited knowledge. The manuscript is concise and clearly written and will be of interest to a broad scientific community.

The previous criticism is addressed constructively.

A minor correction:

1. Fig S5 is the only place where NodT protein is mentioned in the results. Would be helpful to if some explanations are provided.

Our response:

We apologize for this remaining point. In the literature, NodT is also referred to TolC. We now replaced NodT by TolC throughout revised version.

Reviewer #3 (Remarks to the Author):

The authors have fully addressed my previous comments. This crystal structure was critical to interpret the cryo-EM reconstruction of the MacA-MacB-ToIC complex that was recently published in Nature Microbiology. I recommend publication without any delay.

Our response:

I appreciate the recommendation of this reviewer. We also hope that this paper will be published as soon as possible.

Reviewer #4 (Remarks to the Author):

The authors have sensibly removed the undue speculation from their previous version of the manuscript. I have only two points left:

I would like to see the PDB validation report, for quality check on the crystal structure.

Our response:

Shortly after the first submission of our manuscript, we did send the PDB validation report in response to a request from the Editor so that the report could be made available to the reviewers. The data in the report suggest that our MacB structure is of sufficient quality.

-The authors should provide an omit map for figure 2b.

Our response:

Thank you for your useful advice. We included both the experimental map and Fo-Fc omit map as Supplementary Fig. 4 and Fig. 4, respectively.

REVIEWERS' COMMENTS:

Reviewer #4 (Remarks to the Author):

The authors have addressed my comments appropriately.